# Neurorehabilitation through Hippotherapy on Neurofunctional Sequels of Stroke: Effect on Patients’ Functional Independence, Sensorimotor/Cognitive Capacities and Quality of Life, and the Quality of Life of Their Caregivers—A Study Protocol

**DOI:** 10.3390/brainsci12050619

**Published:** 2022-05-09

**Authors:** Hélène Viruega, Carole Imbernon, Nicolas Chausson, Tony Altarcha, Manvel Aghasaryan, Djibril Soumah, Edwige Lescieux, Constance Flamand-Roze, Olivier Simon, Arnaud Bedin, Didier Smadja, Manuel Gaviria

**Affiliations:** 1Institut Equiphoria, 48500 La Canourgue, France; helene.viruega@equiphoria.com; 2Clinical Neurosciences, Alliance Equiphoria, 48500 La Canourgue, France; 3Service de Neurologie et Unité Neurovasculaire, Centre Hospitalier Sud Francilien, 91000 Corbeil-Essonnes, France; carole.imbernon@chsf.fr (C.I.); nicolas.chausson@chsf.fr (N.C.); tony.altarcha@chsf.fr (T.A.); manvel.aghasaryan@chsf.fr (M.A.); djibril.soumah@chsf.fr (D.S.); edwige.lescieux@chsf.fr (E.L.); constance.flamand.roze@gmail.com (C.F.-R.); didier.smadja@chsf.fr (D.S.); 4Boehringer Ingelheim Human Health, 100-104 Avenue de France, 75013 Paris, France; olivier.simon@boehringer-ingelheim.com (O.S.); arnaud.bedin@boehringer-ingelheim.com (A.B.)

**Keywords:** cerebrovascular accident, hippotherapy, neurorehabilitation, silent neurofunctional barriers, functional deficit, neuroplasticity, autonomy, quality of life

## Abstract

Background: Stroke is a high burden illness and the second leading cause of worldwide disability with generally poor recovery rates. Robust benefits of hippotherapy, a novel neurorehabilitation approach, in functional recovery following various severe neurological disabling conditions has been shown. In the present study, we will analyze the effect of a hippotherapy program on the outcome of post-stroke patients in the first year post-stroke. Method: A randomized controlled clinical trial on the effectiveness of hippotherapy (4 weeks/18 weeks hippotherapy/conventional neurorehabilitation) versus conventional neurorehabilitation alone (22 weeks) will be conducted over 48 weeks. In the treated group, one-hour daily hippotherapy sessions will be exclusively conducted during the hippotherapy’s cycles, alternated with periods of conventional neurorehabilitation. A test battery will measure both the functional and psychological outcomes. The primary endpoint will be the patient’s functional independence. The secondary endpoints will measure the sensorimotor function, autonomy, and quality of life, as well as the caregivers’ quality of life. Results and conclusion: Individual brain connectome, life history and personality construct influence the brain’s functional connectivity and are central to developing optimal tailored neurorehabilitation strategies. According to our current practice, hippotherapy allows the enhancement of substantial neuroplastic changes in the injured brain with significant neurological recovery. The protocol aims to confirm those issues. Trial registration in ClinicalTrials.gov NCT04759326 accessed on 19 February 2021.

## 1. Introduction

Nearly 1.1 M Europeans suffer a stroke each year (17 M worldwide), which adds to 33 M stroke survivors [1]. Mortality is about 15% by 1 month, 25% by 1 year, and 50% by 5 years. A third of survivors experience disability (modified Rankin Scale scores 3–5) five years after having a stroke, due to physical, cognitive, and/or emotional deficits [2,3]. Much of the scientific efforts focus on medical programs involving acute care and inpatient rehabilitation. However, long-term disability often occurs and patients may continue to experience serious daily difficulties.

Outpatient neurorehabilitation programs are quite dissimilar, ranging from specialized interdisciplinary programs (involving different configurations of therapies) to limited individual therapies [4]. The need for better post-stroke care is still an issue. Indeed, the late rehabilitative phase is essential to both patients and their caregivers. Patients must receive a current continuum program instead of patchy modules of care shaped by a specific purpose with separate accountability [5]. Additionally, they need to understand the objectives of each phase of post-acute rehabilitation in different contexts and comprehend the challenges encountered at each level for the development of their empowerment and the appropriation of their future.

The notion of reaching a plateau in stroke recovery is questioned by a growing number of studies [6]. Accordingly, there is a need to design and assess therapeutic interventions in neurorehabilitation that promote sustained recovery from stroke over time [7,8]. Multidisciplinary programs need to be developed to cover the complex stroke continuum with the ultimate goal of reducing its large burden [7]. There is growing evidence that a multidisciplinary and cross-sectoral approach is pivotal to allow optimal post-stroke functional recovery [9,10].

Hippotherapy is an emerging specialized rehabilitation approach, performed on a specially trained horse via its movement at a walk, by a team of accredited health professionals (e.g., physicians, psychologists, physical therapists, occupational therapists, psychomotor therapists, and chiropractors) and equine specialists to guide the horse [11,12,13,14,15,16,17,18,19,20,21]. The horse movement at a walk is biomechanically similar to the movement of a human when walking and yields to comparable micro-adjustments of the patient’s postural muscles during hippotherapy [22]. Additionally, through multimodal inputs (sensory, exteroceptive, proprioceptive, interoceptive, and emotional), hippotherapy has a direct action on the individual’s motor capacities, as well as on his/her cognitive abilities [11,12,16,23,24,25,26,27]. During hippotherapy, the specific execution and repetition of a task are key elements of learning/strengthening/promoting a function and a robust neurorehabilitation backbone [11,12,13,14,15,16,17,18,19,20,21,23,24,25,26,27].

Hippotherapy incorporates motor, sensory, cognitive, and psychological stimulation with the goal of promoting recovery after stroke. It has been reported that emotionally empowering multimodal interventions, such as hippotherapy, can provide patients in a late post-stroke phase with life-changing experiences that can have a profound physical and psychological impact [28,29]. In a cohort of patients ranging from 10 months to 5 years post-stroke, hippotherapy showed immediate improvements in functional task and gait performances [29]. Additionally, increasing evidence suggests that rehabilitative activities in stimulating environments can provide functional improvements also in a late phase of recovery [30]. As such, therapeutic horseback riding (≥12 months after stroke) and horse simulator therapy (≥6 months after stroke) improve balance, asymmetric body-weight bearing, motor impairment of lower limbs, independence of ambulation, gait performance, and quality of life, when combined with conventional physiotherapy [31,32,33,34]. These results add to the body of knowledge about how such a program produces objective and subjective changes in stroke survivors, and how it can be used to facilitate the recovery process and enhance self-empowerment after stroke.

Existing standard procedures are not fully suitable for studying complex interventions, such as neurorehabilitation, in a clinical trial. Indeed, this comprehensive approach must be conducted according to specific ethical considerations and it is therefore inappropriate to suspend and/or delay a validated therapy and replace it with an experimental procedure alone, or to provide a true “placebo procedure” in a control group. Therefore, the study will have the following two arms: (i) a treatment arm where patients will receive 4 weeks of hippotherapy care combined with 18 weeks of conventional neurorehabilitation; and (ii) a control arm where patients will receive 22 weeks of conventional neurorehabilitation alone. The primary aim of this clinical trial is to compare the effect of neurorehabilitation through hippotherapy versus conventional care alone on the functional independence of patients in the first year post-stroke. The secondary endpoints will broaden the global evaluation of the patient outcomes and measure the impact of the program on the quality of life and burden of their close caregivers.

## 2. Materials and Methods

### 2.1. Design

We will conduct a randomized open, prospective two-arm controlled trial with blinded clinical evaluations on the effectiveness of hippotherapy (plus conventional neurorehabilitation) versus conventional neurorehabilitation alone. Due to the nature of the clinical trial, participants will know which treatment they will receive and investigators may also be aware of the treatment modality during the patient inclusion/randomization process. In this context, a double- or single-blind procedure is not entirely appropriate/possible. In order to mitigate the effect of factors that are not related to the treatment or intervention being tested, the patient’s functional abilities and emotional sphere will be assessed by two external actors, i.e., two physiatrists associated with the Centre Hospitalier Sud Francilien (see below). These physicians will conducted blinded face-to-face clinical assessments (they have no link with the investigation staff; they do not have access to the whole eCRF; they will also be blinded to previous assessments; and patients and their caregivers will be carefully advised not to disclose the treatment modality they are following). The study will occur at the Equiphoria Institute (La Canourgue, France: https://www.equiphoria.com/en/; accessed on 8 September 2021) for the hippotherapy program and at the Neurology Service of the Centre Hospitalier Sud Francilien (CHSF Corbeil-Essonnes; https://www.chsf.fr/nos-service/medecine/neurologie/; accessed on 8 September 2021) for the conventional neurorehabilitation. Randomization will be stratified by age (18–59 years, 60–74 years, and ≥75 years), stroke type (ischemic versus hemorrhagic), and degree of disability (Rankin scores 3 or 4) to ensure the groups’ homogeneity. The randomization and anonymization of the patient will be performed automatically by an algorithm in the eCRF following the completion of the patient’s initial data at inclusion. The clinical tests will be stored (eCRF) in a secure digital platform of the external subcontractor (CRO) for delayed treatment.

#### 2.1.1. The Study’s Protocol

The protocol’s shape relies on our own clinical experience over the last decade [6,7,8,9,10,11,12,13,35]. The hippotherapy approach will comprise three cycles. The first one consists of two-week daily sessions. This first phase aims to (i) evaluate the patient’s neuro-functional capacities, and (ii) build the initial treatment taking into account the silent barriers that may exist (e.g., PTSD, fatigue, pain, and fear) and that strongly interfere with the individual’s functional outcome [36] developing ways to overcome them. After a nine-week “hippotherapy wash out” period during which the patient continues with the conventional neurorehabilitation care, an intermediate one-week daily hippotherapy cycle followed. Then, after a second nine-week “hippotherapy wash out” period, a final one-week daily hippotherapy cycle will occur. The conventional outpatient rehabilitation for each patient in the hippotherapy group will be temporarily suspended during the hippotherapy cycles. Six months after the end of the rehabilitation protocol, patients will be monitored for sustainability of functional gains. The entire protocol will last 48 weeks (see Figure 1).

The so-called “hippotherapy wash out” period has three main objectives: fixing of the new schemes; translating them to day-to-day skills; and revealing new unmet needs. Our current clinical practice supports the proposed rhythm for rehabilitation through hippotherapy (see Section 2.3.3).

A battery of validated tests are used to study the patient’s level of disability, sensorimotor function, and quality of life. A measurement of the quality of life and burden of caregivers is also included. Patients will undergo evaluations prior and at the end of the neurorehabilitation program, as well as six months after the end of the neurorehabilitation program (end of the study).

#### 2.1.2. Primary Efficacy Endpoint

The Functional Independence Measure (FIM; lasting 30–45 min) (time frame: Changes from baseline to week 22; changes from week 22 to week 48) is an 18-item 7-point Likert scale evaluating physical, psychological, and social function, indicating the amount of assistance required for each task. A final total score is obtained ranging from 18 (complete dependence/total assistance) to 126 (complete independence) [37].

#### 2.1.3. Secondary Efficacy Endpoint

For the patient:

The modified Rankin Scale (mRS; lasting 5–15 min) (time frame: Changes from baseline to week 22; changes from week 22 to week 48) is a standardized measure that describes the extent of disability after stroke. The mRS is a single-item scale. It ranges from 0 (no symptoms) to 6 (death due to stroke) [38,39].

The Fugl-Meyer Assessment (FMA; lasting 30–35 min) (time frame: Changes from baseline to week 22) is a stroke-specific, performance-based impairment/recovery index. It is designed to assess motor functioning, balance, sensation, and joint functioning in stroke patients. Assessment items are scored based on the ability to complete the item using a 3-point Likert scale (0 = cannot accomplish; 1 = partially accomplished; and 2 = completely accomplished) where very severe ranges from 0–35, severe 36–55, moderate 56–79, and light > 79 (max. score 226) [40].

The Berg Balance Scale (BBS; lasting 10–15 min) (time frame: Changes from baseline to week 22) is a 14-item 4-point Likert scale where patients must maintain positions and complete moving tasks of varying difficulty. A global score is calculated out of 56, which represents the ability to independently complete the test [41,42].

The walking distance in 2 min (2-MWT; lasting less than 5 min) (time frame: Changes from baseline to week 22) reports the walking distance of the patient after 2 min. The number and duration of rests during the 2 min are also measured [43].

The Short-Form Health Survey (SF-36; lasting 10 min) (time frame: Changes from baseline to week 22; changes from week 22 to week 48) is a 36-item weighted Likert scale questionnaire, which measures quality of life (QoL) on 8 domains, both physically and emotionally based. The SF-36 is a generic patient-report measure used inter alia in stroke [44].

For the caregiver:

The Short-Form Health Survey (SF-36) for quality of life (time frame: Changes from baseline to week 22)—see above.

The Zarit Burden Inventory (ZBI; lasting 30 min) (time frame: Changes from baseline to week 22) is a 22-item 5-point Likert scale. The higher the score, the more extensive the burden. It is one of the most used instruments to assess caregiving burden in clinical and research settings [45,46].

#### 2.1.4. Sample Size and Recruitment

From the related data [47], the sample-size calculation was based on a minimum reasonably significant difference in the primary outcome (Functional Independence Measure) of 22 points with a standard deviation of σ = 28.43 [48]. To detect that difference with a power of 80% using a two-sample *t*-test at a two-sided significance level of 5%, a total of 44 patients in the randomized arms (22 per arm) will be required. We will recruit 52 patients into the study, to allow for an attrition rate of around 20%.

Recruitment will be conducted during a 12-month period. The study will end when the last patient enrolled undergoes a neurological evaluation to determine the durability of functional recovery.

The end-of-treatment assessment will be completed within two weeks of the end of 22 weeks of treatment and the end-of-protocol assessment will be completed 26 weeks after the end of treatment. The protocol began in the fourth quarter of 2021 and will run through the third quarter of 2024.

### 2.2. Selection/Treatment of Subjects

#### 2.2.1. Participants

Participation in the study will be proposed to every patient hospitalized at the CHSF after a stroke, fulfilling the inclusion criteria. Patients will be drawn at random to build the two arms of the study. The study participants will be adult males or females with a moderate to severe disability (Rankin score ≥ 3 and ≤4 at baseline). Each participant will present written informed consent in accordance with the Declaration of Helsinki. The data will be anonymized and properly stored to respect the confidentiality of sensitive medical data and EU GDPR legislation. Those responsible for the selection and inclusion of patients are the neurologists of the Neurology Service and Neurovascular Unit of the CHSF coordinated by Prof. Didier Smadja, head of department and principal investigator. A certificate of non-contraindication to hippotherapy will be established by the neurologists of the CHSF for each single patient. In case of a patient’s withdrawal, he/she will continue to benefit the planned care with no impact on its quality.

#### 2.2.2. Inclusion and Exclusion Criteria

Inclusion criteria:Age ≥ 18 years old;Ischemic or hemorrhagic acute stroke;Inclusion > 3 and ≤ 12 months post-stroke,Rankin score ≥ 3 and ≤ 4 at inclusion,Written informed consent,Affiliation to social security,Hip minimal abduction of 25 degrees bilateral with no history of hip dislocation and/or dysplasia,Certificate of non-contraindication.

Exclusion criteria:Major cognitive impairment affecting comprehension (Mini Mental State Examination test < 24 points);Global or sensory aphasia;Neurological or psychiatric co-morbidity (other than mild to moderate post-stroke depression);Evidence of uncontrolled seizures;Substance abuse;History of uncontrolled pain;History of allergic reactions to dust and/or horsehair, or severe asthma;Body weight ≥ 110 kg;Contraindications to physical activity;Inability or medical contraindication to travel to the Institute Equiphoria;History of therapeutic horse riding or hippotherapy during the last 6 months;Pregnant or lactating women;Patients participating in other biomedical research or in a period of exclusion.

### 2.3. Interventional Methods

#### 2.3.1. Treated Group

The hippotherapy exercises will consist of four phases performed sequentially:On the horse simulator (Racewood Ltd., Tarporley, UK): (i) 10 min of warm-up allowing patient’s familiarization, muscles’ warm-up, and nervous system facilitation (the Therapeutic Equine Simulator System TESS© has been used as valuable complement for hippotherapy in postural balance rehabilitation [12]); in some cases, the simulator can be used for the entire 1 h session depending on the patient’s needs;On the horse: (ii) 5 min passive and active mobilization of the lower limbs, passive and active stretching of the different muscle groups; (iii) 40 min work on global postural balance and fine-tuning of postural responses (eyes open and closed), work on upper limbs’ fine motor skills by manipulating objects, strengthening of different muscle groups, reinforcement of the body schema and body image [49], breathing techniques and visualizations; and (iv) 5 min relaxation with passive mobilizations and passive stretching, especially of the flexor muscles.

The hippotherapy protocol is substantially the same for every patient knowing that a reinforcement of postural balance, symmetry, hip and shoulder dissociation, spinal joint mobility, and muscle tone regularization are needed in both Rankin 3 and 4 patients. These will be obtained through the horse movement at a walk and the simulator movement. During hippotherapy, the postural balance work and the muscle tone regularization are background tasks [11,12]. The physiotherapist will simultaneously work the upper limb through repetitive task-specific training, muscle strength training, bilateral training, and mental practice. The intensity of the exercises will be fully tailored and will depend on the patient’s clinical heterogeneity, fatigability, and confounding factors.

Standing and walking is not mandatory for mounting a horse during hippotherapy. The institute is equipped with a ramp, which allows for the wheelchair to be positioned alongside the horse. Our trained staff operate the transfers of non-walking patients. Once on the horse, the patient is usually able to maintain an adequate postural balance. The therapist walking alongside the horse during the session is an occasional form of support when needed. The horse is equipped with a specially manufactured leather pad with handles (no saddle, no stirrups), allowing for the patient’s maximum freedom of movement and contact with the horse’s back.

During the remaining 18 weeks, the treatment options for each patient will include physiotherapy (motor training and functional training), occupational therapy, language therapy, and psychological and social support in the CHSF (see Section 2.3.2 below).

#### 2.3.2. Control Group

Patients of the control group will follow a standard outpatient rehabilitation treatment during 22 weeks, consisting of a combination of five half-day physiotherapy, occupational therapy, speech therapy and psychotherapy sessions per week according to the patient’s needs. For the patients with a modified ranking scale comprised between 3 and 4, it is likely that the work targeted on gait, balance, and mobility will focus on gait-oriented physical fitness training, repetitive task training, muscle strength training, and the treadmill training when possible. People with difficulty using their upper limb should be given the opportunity to undertake as much tailored practice of upper limb activity as possible. Interventions, which can be used routinely, involve constraint-induced movement therapy in selected people, repetitive task-specific training, and mechanical-assisted training. One or more of the following interventions can be used in addition to the ones above: mental practice, electrical stimulation, biofeedback in conjunction with conventional therapy, bilateral training, and mirror therapy. Moreover, special attention will be paid to manage mild to moderate spasticity through early comprehensive rehabilitation. Additionally, contractures must be carefully monitored and prevented by conventional motion therapy. Particularly common is the loss of shoulder external rotation, elbow extension, forearm supination, wrist and finger extensions, ankle dorsiflexion, and hip internal rotation. People with severe weakness tend to develop contractures. Any joint or muscle not regularly used can be subject to soft tissue complications, which will eventually limit movement and may cause pain.

As several techniques are likely to be used, a record of the number of rehabilitation sessions (physiotherapy, occupational therapy, speech therapy, and psychotherapy) and their type will be collected for each patient during the study period and used as a covariate.

#### 2.3.3. Intensity and Effectiveness

It is difficult to determine the minimum effective quantity of the required neurological rehabilitation. Scientific evidence is lacking and studies have biases of several kinds. To date, the effectiveness of many rehabilitation techniques has not been systematically demonstrated. According to a 2014 review of the literature [50], the studies carried out are highly heterogeneous (e.g., type of rehabilitation, included patients, quality of structures, and type of evaluation) and have a low methodological quality [50,51,52]. The overall effectiveness of stroke rehabilitation has been substantiated from older studies generally mainly showing reductions in mortality, dependency rates, and risk of institutionalization.

On the one hand, in this rather empirical context, sessions of at least 45 min by type of rehabilitation are commonly recommended (more if the patient can bear it). It is accepted that the frequency is daily or at least 5 days a week. Rehabilitation should be conducted for at least 8 weeks “as long as improvement continues”. During the first two weeks, frequent, short sessions of progressive intensity are preferred. The intensity with which rehabilitation is conducted varies depending on dispensing structures. Thus, rehabilitation in a specialized service with full hospitalization or day hospitalization offers possibilities to perform intensive multiple rehabilitation, which is more difficult to perform with independent professionals [50].

On the other hand, a hippotherapy session lasts one hour per day during which between 3000 and 4500 contractions of each postural muscle are sequentially realized in a background mode (horse at a walk) in parallel to other requests (fine motor skills, cognitive elaboration, and psychic work), well beyond what a conventional rehabilitation session allows. Given the intensity of each session that mobilizes the individual in his/her entirety (e.g., somatic, sensory, cognitive, emotional and motivational, and psychic spheres), and relying on our clinical experience over the last decade, we unequivocally respect a certain rhythm by integrating the duration of the patient’s processing of physical/mental skills and the ensuing fatigue. Additionally, thanks to the enriched environment brought by hippotherapy, we notice continuous functional improvements, even beyond the theoretical period of consolidation of the neurological outcome after injury [53,54]. Overall, the strong stimulation of the sensory, sensitive, and motor spheres promotes and interacts with the mechanisms related to the tasks’ performances in the cognitive and emotional domains through the activation of multiple neural networks [35,55]. The degree of change associated with neuroplasticity through hippotherapy is most likely linked to both the relevance of the activity and the intensity and frequency of the elements that constitute it [56,57,58]. A systematic validation is carried out at the Equiphoria Institute to provide a solid theoretical foundation for this approach.

#### 2.3.4. Safety/Adverse Events

At each evaluation, the investigator will determine whether any adverse events or serious adverse events (AEs or SAEs) occurred. All adverse events occurring during the study will be recorded on the appropriate case report form (eCRF) page by the investigator. The nature, severity, and relation of the adverse event to the study protocol and treatment will be documented. A safety plan has been duly integrated in the Trial Master File.

### 2.4. Data Collection and Analysis

The data will be duly anonymized and blindly manipulated in the eCRF. Data management will be performed by a CRO already contracted according to a data management plan. A data monitoring committee composed by the CRA of the CRO and the principal investigators of the CHSF will oversee the follow-up of the data collection accuracy and reporting of adverse events.

#### 2.4.1. Analysis Population Sets

All analyses will be conducted on the Intention-To-Treat (ITT: all enrolled patients undergoing the neurorehabilitation procedure) and Per-Protocol (PP: all patients completing the study without major protocol deviations) populations. All measured variables and derived parameters will be individually listed and, if appropriate, tabulated by descriptive statistics.

#### 2.4.2. Statistical Analysis

Statistical methods will be in accordance with the CONSORT statement for reporting of randomized trials. Appropriate non-parametric statistics will be used to evaluate the comparability of the intervention and control group at baseline in terms of the clinical characteristics. If, despite the stratified randomization procedure, the groups are not comparable for one or more background variables, those variables will be routinely employed as covariates in the subsequent analyses. The scores for each variable (primary and secondary endpoints) will be calculated according to the published scoring algorithms. Intergroup (effect of the treatment on the measured variable, compared to the control) and intragroup (evolution of the measured variable on the same group) differences in the mean scores over time will be tested using a multilevel analysis. The effect sizes will be calculated using standard statistical procedures. All analyses will be conducted on an ITT basis. PP analyses will also be carried out (as secondary analyses). To test the effectiveness of hippotherapy intervention on functional independence (FIM; primary outcome), non-parametric ANOVA measures for non-linear scores (Friedman’s test for paired data) will be conducted. In addition, to test the effectiveness of the intervention on the secondary outcome variables, including the caregivers’ measurements, non-parametric ANOVA procedures (Friedman’s test for paired data) will also be applied. Post hoc analyses (e.g., additional correlation and covariance statistical procedures) will be conducted according to the obtained results. They will fine-tune the initial data analysis by taking into account potential confounding factors. Given the power of the study (80%), the post hoc results will be statistically robust and therefore reliable. The threshold of statistical significance will be *p* = 0.05. The statistical analyses will be implemented by the CRO using SAS^®^ software (version 9.4, SAS Institute, Cary, NC, USA).

## 3. Discussion

Scientific evidence has gradually emerged in recent years, showing the benefits of hippotherapy in various severe brain disabling conditions, including stroke [11,12,13,14,15,16,17,18,19,20,21,23,24,25,26,27,59]. An increasing number of studies on the hippotherapy treatment of neurological sequelae of acquired brain injuries are published, but their methodological quality is rather low. In general, they report the beneficial effects on posture, walking, spasticity, and gross and fine motor skills, but these positive effects are challenged by the difficulties in establishing sound methodological frameworks. The present protocol is the result of a decade of clinical hands-on practice and research. The rationale of this cutting-edge method considers motion as a key driver of short- and long-term neurorehabilitation strategies. It could be hypothesized that movement could promote a reshaping of brain functions, thanks to the up/down regulation of the synthesis and release of neuroactive substances (e.g., BDNF, VIP, GABA, endomorphin 1 and 2, α-MSH, serotonin, and dopamine) synthesized in the enteric nervous system, which would physiologically impact the entire cerebral metabolism fostering neuroplasticity [27,60,61].

Each horse conveys a specific movement at a walk [22,62,63]. The therapeutic strategy here consists of thoroughly evaluating the patient’s condition and deciding what movement is most suitable to release the potential psychic/cognitive (silent patient’s blockages) and functional (trunk hypertonia, inefficient postural balance, postural tilt, pelvic locking, shoulder asymmetry, attentional lability, executive dysfunctions, and deficit in spatial awareness) barriers. By identifying the very first step of the process, the therapist would be able to trigger the sequence and progressively unlock the neuro-functional capacities (e.g., trunk motion and modulation of postural reflexes through the setup of the alternating contraction of postural muscles, depending on micro-movements elicited by the horse at a walk). Simultaneously, the reinforcement of the cognitive sphere (e.g., executive functions, attention, and working memory) could be efficiently targeted and tailor-made. Some of the clinical hypotheses on which this project is built have been strengthened and confirmed through several quantitative research protocols on postural kinetics and electromyography [11,12].

Hippotherapy gradually emerges as a cutting-edge solution for human impairment, activity limitation, and participation restriction. The main objective of the clinical trial, which relies on the protocol presented in the present study, will be to validate hippotherapy as a novel approach to stroke neurorehabilitation consisting of a personalized all-inclusive rhythm-adapted patient’s empowerment, which constantly re-evaluates and adjusts itself in real time, according to the patient’s neurological possibilities, his/her interaction with the family caregiver(s) and his/her lifetime goals.

## 4. Conclusions

Given the highly specific intervention and setting around hippotherapy, some elements regarding applicability, cost effectiveness, and generalizability are discussed below.

Currently, the end of formal inpatient neurorehabilitation (frequently by 6 months after having a stroke) should not mean the end of the rehabilitation processes. In many respects, stroke has been medically managed as a temporary or transient condition, instead of a chronic condition that requires monitoring after the acute and sub-acute phases. The unmet needs persist in many domains, including social reintegration, health-related quality of life, maintenance of activity, and self-efficacy (i.e., belief in one’s capability to express a behavior). Apathy is manifested in >50% of survivors at 1 year after stroke [64]; fatigue is a common and debilitating symptom in chronic stroke [65]; the daily physical activity of community-living stroke survivors is low [66]; and depressive symptomology is high [67]. By 4 years after onset, >30% of stroke survivors report persistent participation restrictions (e.g., difficulty with autonomy, engagement, or fulfilling societal roles) [68].

The hippotherapy approach combines (i) a multidisciplinary cross-sectoral team simultaneously around the patient, (ii) a global approach focused on the individual and his/her potentials rather than on the disabilities, and (iii) strict care protocols that include a program for caregivers. The program provides an innovative disruptive response to the conventional care of diseases of neurological origin. In the context of conventional compartmentalized marginally effective neurorehabilitation approaches for the stroke improvement of patients globally, the final objective will be that such a hippotherapy program will be rapidly adopted as a “gold standard” neurorehabilitation alternative, and hence included in the stroke medical guidelines.

An independent agency (part of the Deloitte Group, 2017 internal report) has been mandated to study whether hippotherapy’s programs generated reduced costs for the nation’s health system. The financial costs associated with stroke care are high. In France, for example, the total healthcare cost for stroke was EUR 5.3 billion in 2007, including nursing care, hospitalization, and medicines, covered almost entirely by the National Social Security. Each new stroke costs EUR 17,000–20,000 in the first year, while follow-up care for survivors costs on average EUR 8000 per year [69,70]. For each stroke patient that attends a hippotherapy program, such as ours, the cost avoided is between EUR 19,000 up to EUR 70,000, depending on the level of autonomy of the patient (4 levels have been defined). Multiplied by the number of patients who can potentially be treated, the spared spending for the health system globally amounts to millions each year.

From the economical point of view, one of the obstacles that can be encountered is the fact that neurorehabilitation through hippotherapy is considered as an additional non-refundable cost. Nevertheless, the positive results obtained with this approach encourage French national insurance companies to reimburse up to 100% of the program costs to patients. From the point of view of medical recognition, a slow acceptance might occur as the scientific community of reference (neurologists and neuro-rehabilitators) are not familiar with hippotherapy protocols and methodologies. They may thus be skeptical and low adopters. Significant efforts must be devoted for an effective dissemination among key opinion leaders and the scientific and hospital communities. In addition to publications, the program’s results will be presented at international events, and the staff will participate in working groups to inform clinicians, patient associations, and insurance companies about the approach, and thus optimize its generalizability.

## Figures and Tables

**Figure 1 brainsci-12-00619-f001:**
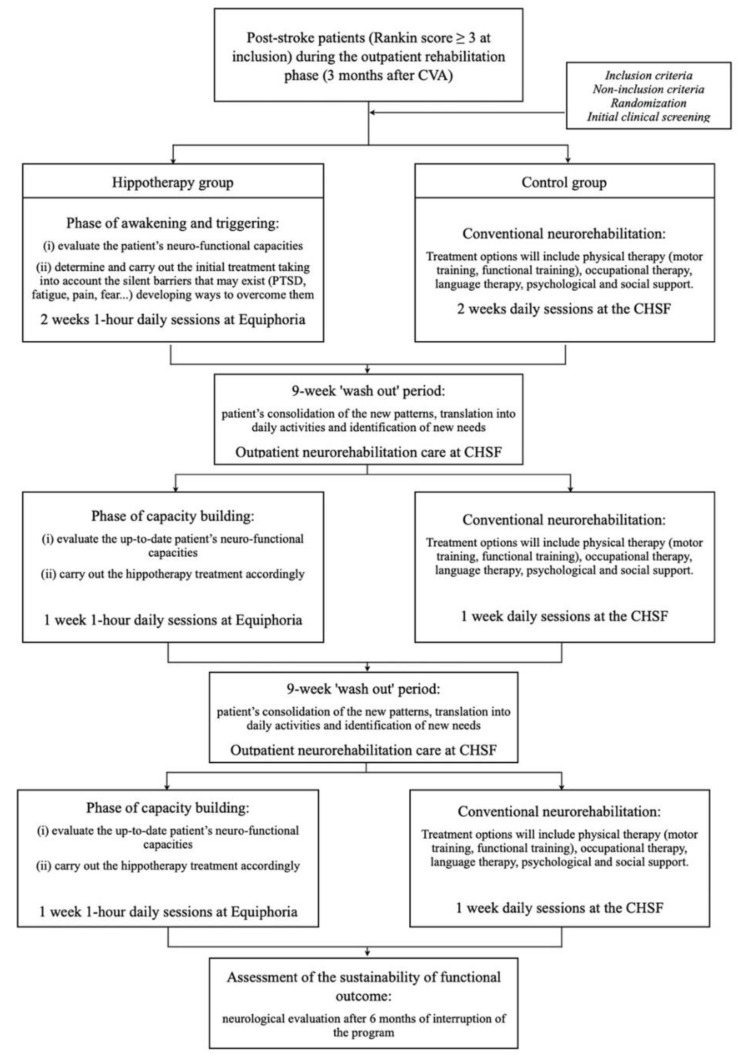
Protocol’s scheme.

## Data Availability

The datasets generated during the current study will be made available from the corresponding authors on reasonable request.

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
