# Peer review of "Neurorehabilitation through Hippotherapy on Neurofunctional Sequels of Stroke: Effect on Patients’ Functional Independence, Sensorimotor/Cognitive Capacities and Quality of Life, and the Quality of Life of Their Caregivers—A Study Protocol"

_brainsci, 2022, doi:10.3390/brainsci12050619_

Round 1

Reviewer 1 Report

The manuscript depicts a novel topic regarding post-stroke rehabilitation using hippotherapy, as a study protocol.

Some issues must be addressed by the authors:

In the introduction section, the authors must provide detailed information regarding hippotherapy- recent studies- regarding how this type of therapy influence human body mechanics and posture- with specific details regarding also proprioception and feedback. Also, since the future study is intended to investigate hippotherapy in the subacute-to chronic phase of post-stroke patients, further details shall be provided regarding the post-stroke stages and the importance of neuroplasticity. Also, from the introduction aim, it seems that the research will have 2 different arms, in the further section of the manuscript is described the fact that both groups will receive at a point from the research- rehabilitation services. Therefore, the aim should be more appropriately described.

In the material and methods section:

-the authors should clearly describe how the trial will be blinded
-what means dedicated procedures?
-The authors should describe the hippotherapy and conventional physiotherapy protocols which intend to use, specific, and based on their stratified randomization and also on assessments criterion (ex: for patients with FIM > 80 and <90, the hippotherapy and physiotherapy protocol will consist of...)

Also regarding the analysis, the authors must take into account the subacute and the chronic stages of post-stroke patients since this can bias the research results.

Also, the authors must discuss and identify the costs of this therapy versus the potential benefits.

Author Response

Thank you for your comments that allowed us to improve the quality of our paper.

The manuscript depicts a novel topic regarding post-stroke rehabilitation using hippotherapy, as a study protocol.

Some issues must be addressed by the authors:

In the introduction section, the authors must provide detailed information regarding hippotherapy- recent studies- regarding how this type of therapy influence human body mechanics and posture- with specific details regarding also proprioception and feedback.

The following details were added to support the effect of this type of therapy on human body mechanics and posture (lines 75-89):

Hippotherapy incorporates motor, sensory, cognitive and psychological stimulation with the goal of promoting recovery after stroke. It has been reported that emotionally empowering multimodal interventions, such as hippotherapy, can provide patients in a late post-stroke phase with life-changing experiences that can have a profound physical and psychological impact [29-30]. In a cohort of patients ranging from 10 months to 5 years post-stroke, hippotherapy showed immediate gains in functional task performance and gait performance [30]. Also, increasing evidence suggests that rehabilitative activities in stimulating environments can provide functional improvements also in a late phase of recovery [31]. As such, therapeutic horseback riding (≥12 months after stroke) and horse simulator therapy (≥6 months after stroke) improve balance, asymmetric body weight bearing, motor impairment of lower limbs, independence of ambulation, gait performance, and quality of life, when combined with conventional physiotherapy [32-35]. These results add to the body of knowledge about how such a program produces objective and subjective changes in stroke survivors and how it can be used to facilitate the recovery process and enhance self-empowerment after stroke.

References:

  1. Bunketorp-Käll, L.; Lundgren-Nilsson, Å.; Samuelsson, H.; Pekny, T.; Blomvé, K.; Pekna, M.; Pekny, M.; Blomstrand, C.; Nilsson, M. Long-Term Improvements After Multimodal Rehabilitation in Late Phase After Stroke: A Randomized Controlled Trial. Stroke 2017, 48(7), 1916-1924. doi: 10.1161/STROKEAHA.116.016433.
  2. Bunketorp-Käll, L.; Pekna, M.; Pekny, M.; Blomstrand, C.; Nilsson, M. Effects of horse-riding therapy and rhythm and music-based therapy on functional mobility in late phase after stroke. NeuroRehabilitation 2019, 45(4), 483-492. doi: 10.3233/NRE-192905.
  3. Pohl, P.; Carlsson, G.; Bunketorp-Käll, L.; Nilsson, M.; Blomstrand, C. A qualitative exploration of post-acute stroke participants' experiences of a multimodal intervention incorporating horseback riding. PLoS One 2018, 13(9), e0203933. doi: 10.1371/journal.pone.0203933.
  4. Beinotti, F.; Correia, N.; Christofoletti, G.; Borges, G. Use of hippotherapy in gait training for hemiparetic post-stroke. Arq. Neuropsiquiatr. 2010, 68(6), 908-913. doi: 10.1590/s0004-282x2010000600015.
  5. Beinotti, F.; Christofoletti, G.; Correia, N.; Borges, G. Effects of horseback riding therapy on quality of life in patients post stroke. Top Stroke Rehabil. 2013, 20(3), 226-232. doi: 10.1310/tsr2003-226.
  6. Sung, Y.H.; Kim, C.J.; Yu, B.K.; Kim, K.M. A hippotherapy simulator is effective to shift weight bearing toward the affected side during gait in patients with stroke. NeuroRehabilitation 2013, 33(3), 407-412. doi: 10.3233/NRE-130971.
  7. Kim, Y.N.; Lee, D.K. Effects of horse-riding exercise on balance, gait, and activities of daily living in stroke patients. J. Phys. Ther. Sci. 2015, 27(3), 607-609. doi: 10.1589/jpts.27.607.

Also, since the future study is intended to investigate hippotherapy in the subacute-to-chronic phase of post-stroke patients, further details shall be provided regarding the post-stroke stages and the importance of neuroplasticity.

The following details were added to support the subacute-to-chronic intervention (lines 57-63):

The notion of reaching a plateau in stroke recovery is questioned by a growing number of studies [6]. Accordingly, there is a need to design and assess therapeutic interventions in neurorehabilitation that promote sustained recovery from stroke over time [7-8]. Multidisciplinary programs need to be developed to cover the complex stroke continuum with the ultimate goal of reducing its massive burden [9]. There is growing evidence that a multidisciplinary and cross-sectoral approach is pivotal to allow optimal post-stroke functional recovery [10-11].

References:

  1. Demain, S.; Wiles, R.; Roberts, L.; McPherson, K. Recovery plateau following stroke: fact or fiction? Disabil. Rehabil. 2006, 28(13-14), 815-821. doi: 10.1080/09638280500534796.
  2. Bernhardt, J.; Borschmann, K.; Boyd, L.; Carmichael, S.T.; Corbett, D.; Cramer, S.C.; Hoffmann, T.; Kwakkel, G.; Savitz, S.; Saposnik, G.; et al. Moving Rehabilitation Research Forward: Developing Consensus Statements for Rehabilitation and Recovery Research. Neurorehabil. Neural Repair 2017, 31(8), 694-698. doi: 10.1177/1545968317724290.
  3. Bernhardt, J.; Hayward, K.S.; Kwakkel, G.; Ward, N.S.; Wolf, S.L.; Borschmann, K.; Krakauer, J.W.; Boyd, L.A.; Carmichael, S.T.; Corbett, D.; et al. Agreed definitions and a shared vision for new standards in stroke recovery research: The Stroke Recovery and Rehabilitation Roundtable taskforce. Int. J. Stroke 2017, 12(5), 444-450. doi: 10.1177/1747493017711816.
  4. Bernhardt, J.; Borschmann, K.; Boyd, L.; Carmichael, S.T.; Corbett, D.; Cramer, S.C.; Hoffmann, T.; Kwakkel, G.; Savitz, S.; Saposnik, G.; et al. Moving Rehabilitation Research Forward: Developing Consensus Statements for Rehabilitation and Recovery Research. Neurorehabil. Neural Repair 2017, 31(8), 694-698. doi: 10.1177/1545968317724290.
  5. Corbett, D.; Jeffers, M.; Nguemeni, C.; Gomez-Smith, M.; Livingston-Thomas, J. Lost in translation: rethinking approaches to stroke recovery. Prog. Brain Res. 2015, 218, 413-434. doi: 10.1016/bs.pbr.2014.12.002.
  6. Malá, H. ; Rasmussen, C.P. The effect of combined therapies on recovery after acquired brain injury: Systematic review of preclinical studies combining enriched environment, exercise, or task-specific training with other therapies. Restor. Neurol. Neurosci. 2017, 35(1), 25-64. doi: 10.3233/RNN-160682.

Also, from the introduction aim, it seems that the research will have 2 different arms, in the further section of the manuscript is described the fact that both groups will receive at a point from the research- rehabilitation services. Therefore, the aim should be more appropriately described.

The following details were added to better describe the two-arms trial in the introduction (lines 90-105):

Existing standard procedures are not fully suitable for studying complex interventions such as neurorehabilitation in a clinical trial. Indeed, this comprehensive approach must be conducted according to specific ethical considerations and it is therefore inappropriate to suspend and/or delay a validated therapy and replace it with an experimental procedure alone, or to provide a true "placebo procedure" in a control group. Therefore, the study will have the following two arms: (i) a treatment arm where patients will receive 4 weeks of hippotherapy care combined with 18 weeks of conventional neurorehabilitation; and (ii) a control arm where patients will receive 22 weeks of conventional neurorehabilitation alone.

In the material and methods section:

  • The authors should clearly describe how the trial will be blinded

We have added the following sentence concerning how the trial will be blinded (lines 114-124) and remove redundant information (from line 160):

Due to the nature of the clinical trial, participants will know which treatment they will receive and investigators may also be aware of the treatment modality during the patient inclusion/randomization process. In this context, a double- or single-blind procedure is not entirely appropriate/possible. In order to mitigate the effect of factors that are not related to the treatment or intervention being tested, the patient's functional abilities and emotional sphere will be assessed by two external actors, i.e. two physiatrists associated with the Centre Hospitalier Sud Francilien (see below). These physicians will carry out blinded face-to-face clinical assessments (they have no link with the investigation staff; they do not have access to the whole eCRF; they will also be blinded to previous assessments; patients and their caregivers will be carefully advised not to disclose the treatment modality they are following).

  • What means dedicated procedures?

The sentence was reformulated to clarify (lines 131-133):

Randomization and anonymization of the patient will be performed automatically in the eCRF by an algorithm following the completion of the patient's initial data at inclusion.

  • The authors should describe the hippotherapy and conventional physiotherapy protocols which intend to use, specific, and based on their stratified randomization and also on assessments criterion (ex: for patients with FIM > 80 and <90, the hippotherapy and physiotherapy protocol will consist of...)

We have added the following descriptions concerning conventional physiotherapy (lines 341-357) and hippotherapy (lines 317-325) protocols:

Concerning the conventional physiotherapy protocols, only general information about the protocol of care can be provided given the myriad of configurations of the neurological deficits:

In patients with a modified ranking scale comprised between 3 and 4, it is likely that the work targeted on gait, balance and mobility will focus gait-oriented physical fitness training, repetitive task training, muscle strength training, and treadmill training when possible. People with difficulty using their upper limb should be given the opportunity to undertake as much tailored practice of upper limb activity as possible. Interventions which can be used routinely involve constraint induced movement therapy in selected people, repetitive task-specific training, mechanical assisted training. One or more of the following interventions can be used in addition to the ones above: mental practice, electrical stimulation, biofeedback in conjunction with conventional therapy, bilateral training, mirror therapy. Moreover, special attention will be given to manage mild to moderate spasticity through early comprehensive rehabilitation. Also, contractures must be carefully monitored and prevented by conventional motion therapy. Particularly common is loss of shoulder external rotation, elbow extension, forearm supination, wrist and finger extension, ankle dorsiflexion and hip internal rotation. People with severe weakness tend to develop contractures. Any joint or muscle not used regularly can be subject to soft tissue complications which eventually will limit movement and may cause pain.

With respect to hippotherapy:

The hippotherapy protocol is substantially the same for every patient knowing that a reinforcement of postural balance, symmetry, hip and shoulder dissociation, spinal joint mobility, and muscle tone regularization are needed in both Rankin 3 and 4 patients. These will be obtained through the horse movement at a walk and the simulator movement. During hippotherapy, the postural balance work and the muscle tone regularization are background tasks [12-13]. The physiotherapist will work simultaneously the upper limb through repetitive task-specific training, muscle strength training, bilateral training, and mental practice. The intensity of the exercises will be fully tailored and will depend on patient’s clinical heterogeneity, fatigability and confounding factors.

Also regarding the analysis, the authors must take into account the subacute and the chronic stages of post-stroke patients since this can bias the research results.

We have added the following descriptions to the paragraph 2.4.2. Statistical analysis (lines 445-449):

Post hoc analyses (e.g., correlation and covariance statistical procedures) will be done according to the obtained results. They will fine-tune the initial data analysis by taking into account potential confounding factors. Given the power of the study (80%), the post hoc results will be statistically robust and therefore reliable.

Also, the authors must discuss and identify the costs of this therapy versus the potential benefits.

The landscape of therapy cost and benefits is provided in Conclusion (it would be improper to talk directly about the costs of the therapy since it could be misinterpreted as a promotional paper for the institute). We prefer to provide factual data of reimbursement by French health insurance companies (see lines 547-551) and we have added some additional impartial financial information (lines 538-544):

Financial costs associated with stroke care are huge. In France for example, the total healthcare cost of stroke was €5.3 billion in 2007, including nursing care, hospitalization and medicines, almost entirely covered by the National Social Security. Each new stroke costs 17k€ - 20k€ in the first year, while follow up care for survivors cost on average 8k€ per year [70-71]. For each stroke patient that attend a hippotherapy program like ours, the cost avoided is between 19k€ up to 70k€, depending on the level of autonomy of the patient (4 levels had been defined).

References:

  1. Chevreul, K.; Durand-Zaleski, I.; Gouépo, A.; Fery-Lemonnier, E.; Hommel, M.; Woimant, F. Cost of stroke in France. Eur. J. Neurol. 2013, 20(7), 1094-1100. doi: 10.1111/ene.12143.
  2. Schmidt, A.; Heroum, C.; Caumette, D.; Le Lay, K.; Bénard, S. Acute Ischemic Stroke (AIS) patient management in French stroke units and impact estimation of thrombolysis on care pathways and associated costs. Cerebrovasc. Dis. 2015, 39(2), 94-101. doi: 10.1159/000369525.

Reviewer 2 Report

  • The paper looks like a concept project that is supposed to promote Institute Equiphoria.... 
  • Lack of evidence related to the case study or pilot study.
  • Why the Authors use "will be"?
  • Biomechanical descriptions should be carefully described and given related to the: horse simulator and alive horse used for hippotherapy
  • The Authors should specify which (simulator or alive horse) they are planning to use and the range of application (biomechanical factors)
  • Study's protocol looks like a concept. The current presentation is confusing. All components (also Time)  should be specified.
  • Sample size and recruitment are very messy. Some chunks related to the Authors previous study are given. But these chunks are given without any background
  • "2.4.2. Statistical analysis" looks like a concept that the Authors are planning to use
  • In discussion the Authors claim that “The main objective here is to validate this neurorehabilitation program consisting in a personalized all-inclusive rhythm-adapted  patient’s empowerment…”  the paper  not contain any validation data!….

Author Response

Thank you for your comments that allowed us to improve the quality of our paper.

The paper looks like a concept project that is supposed to promote Institute Equiphoria....

We thank the reviewer for her/his advice. It was not our intention to promote our institute but to share our thoughts on the tools available for stroke neurorehabilitation. We have rephrased the sentences that could be misunderstood.

Lack of evidence related to the case study or pilot study.

We have not published up today case studies or pilot studies concerning hippotherapy and stroke. Conversely, we have worked and published on hippotherapy and traumatic brain injury, and hippotherapy and cerebral palsy. Given the positive results presented in the mentioned papers (see References [12-14]) and taking into account our clinical experience during ten years in post-stroke patients, we considered relevant to design the present protocol.

Evidence of hippotherapy efficacy in stroke patients is provided in the new added paragraphs in Introduction (besides the corresponding references [6-11,29-35]).

Why the Authors use "will be"?

We use “will be” because it is the protocol of the clinical trial and no patients have been recruited yet. Consequently, the different actions to be taken are in the near future.

Biomechanical descriptions should be carefully described and given related to the: horse simulator and alive horse used for hippotherapy

Biomechanics of horse movement (and simulator movement) are beyond the scope of this work.

The Authors should specify which (simulator or alive horse) they are planning to use and the range of application (biomechanical factors)

The protocol describes the use of both (horse and simulator) sequentially during each of the 20 sessions in the hippotherapy treated group (see lines 302-325). Biomechanical factors are not directly targeted in this study. The potential improvement of gross and fine motor function will be analyzed through neurological scales and would give us the functional status of the patients.

Study's protocol looks like a concept. The current presentation is confusing. All components (also Time) should be specified.

The study protocol has been improved (see lines 317-325 and 341-357). Nevertheless, the descriptions concerning conventional physiotherapy and hippotherapy protocols remain slightly generic given the myriad of configurations of the neurological deficits that we will face. We cannot at this stage standardize a neurorehabilitation protocol, we just can give a general framework of intervention. However, since patients to be recruited are ranged from 3 to 4 in the modified Ranking Scale (3: moderate disability requiring some help, but able to walk without assistance; and 4: moderately severe disability, unable to walk and attend to bodily needs without assistance), the described exercises will be used accordingly during the follow up period.

Sample size and recruitment are very messy. Some chunks related to the Authors previous study are given. But these chunks are given without any background

The paragraph has been rewritten (lines 211-216):

From related data [48], the sample-size calculation was based on a minimum reasonably significant difference in primary outcome (Functional Independence Measure) of 22 points with a standard deviation of σ = 28.43 [49]. To detect that difference with a power of 80% using a two-sample t-test at a two-sided significance level of 5%, a total of 44 patients in the randomized arms (22 per arm) will be required. We will recruit 52 patients into the study, to allow for an attrition rate of around 20%.

"2.4.2. Statistical analysis" looks like a concept that the Authors are planning to use

The statistical analysis description corresponds indeed to the framework that we will use at the end of the clinical trial, when the whole data will be collected (as a reminder, this is only the protocol of the clinical trial). However, we tried to be more specific in the description of the statistical methodology. The paragraph has been rephrased accordingly (lines 429-451):

Statistical methods will be in accordance to the CONSORT statement for reporting of randomized trials. Appropriate non-parametric statistics will be used to evaluate the comparability of the intervention and control group at baseline in terms of clinical characteristics. If, despite the stratified randomization procedure, the groups are not comparable on one or more background variables, those variables will be employed routinely as covariates in subsequent analyses. Scores for each variable (primary and secondary end points) will be calculated according to published scoring algorithms. Intergroup (effect of treatment on the measured variable compared to control) and intragroup (evolution of the measured variable on the same group) differences in mean scores over time will be tested using multilevel analysis. Effect sizes will be calculated using standard statistical procedures. All analyses will be conducted on ITT basis. PP analyses will also be carried out (as secondary analyses). To test the effectiveness of hippotherapy intervention on functional independence (FIM; primary outcome), non-parametric ANOVA measures for non-linear scores (Friedman's test for paired data) will be carried out. In addition, to test the effectiveness of the intervention on the secondary outcome variables, including caregivers’ measurements, non-parametric ANOVA procedures (Friedman's test for paired data) will be also applied. Post hoc analyses (e.g., additional correlation and covariance statistical procedures) will be done according to the obtained results. They will fine-tune the initial data analysis by taking into account potential confounding factors. Given the power of the study (80%), the post hoc results will be statistically robust and therefore reliable. The threshold of statistical significance will be P = 0.05. The statistical analyses will be implemented by the CRO using SAS® software (version 9.4, SAS Institute, NC, Cary, USA).

In discussion the Authors claim that “The main objective here is to validate this neurorehabilitation program consisting in a personalized all-inclusive rhythm- adapted patient’s empowerment...” the paper not contain any validation data!....

The sentence has been reworded to avoid any misinterpretation. Indeed, the use of the present “is to validate” is inappropriate (lines 507-512):

The main objective of the clinical trial, which relies on the protocol presented here, will be to validate hippotherapy as a novel neurorehabilitation approach consisting of a personalized all-inclusive rhythm-adapted patient’s empowerment, that constantly re-evaluates and adjusts in real time, according to the patient’s neurological possibilities, his/her interaction with family caregiver(s) and lifetime goals.

Round 2

Reviewer 1 Report

The authors performed the requested adjustments, but I just have to make some minor suggestions:

In lines 342-357, the authors should mention how they will consider multiple variables for the analysis since multiple rehabilitation or physiotherapy methods or techniques will be used. The authors should carefully be aware of the heterogeneity of their research study and therefore to publish this protocol for the following research, the complex methodology must be based upon. Perhaps the authors are considering a pilot study before the trial.

Author Response

In lines 342-357, the authors should mention how they will consider multiple variables for the analysis since multiple rehabilitation or physiotherapy methods or techniques will be used. The authors should carefully be aware of the heterogeneity of their research study and therefore to publish this protocol for the following research, the complex methodology must be based upon. Perhaps the authors are considering a pilot study before the trial.

Dear Colleague,

Thank you for your comment. Indeed, this potential heterogeneity has been considered in our methodology. We have added the following paragraph on lines 358-361 to clarify it accordingly:

"As several techniques are likely to be used, a record of the number of rehabilitation sessions (physiotherapy, occupational therapy, speech therapy, psychotherapy...) and their type will be collected for each patient during the study period and used as a covariate."